# Clinical and Forensic Aspects of the Management of Child Abuse: The Experience of the Paediatric Emergency Department in Novara, North-West Italy

**DOI:** 10.3390/ijerph20032028

**Published:** 2023-01-22

**Authors:** Micol Puppi, Linda Rota, Lorenza Scotti, Ivana Rabbone, Sarah Gino

**Affiliations:** 1School of Medicine, University of Eastern Piedmont, Via Solaroli 17, 28100 Novara, Italy; 2Division of Paediatrics, Department of Health Sciences, University of Eastern Piedmont, Via Solaroli 17, 28100 Novara, Italy; 3Department of Translational Medicine, University of Eastern Piedmont, Via Solaroli 17, 28100 Novara, Italy; 4Department of Health Sciences, University of Eastern Piedmont, Via Solaroli 17, 28100 Novara, Italy

**Keywords:** violence against minor, maltreatment, sexual violence, peer violence, COVID-19

## Abstract

Background: Child abuse is an endemic phenomenon that refers to any form of violence aimed at children and adolescents. The Emergency Room is often the entry point to healthcare for the abused child. Methods: This is a cross-sectional study including minors, aged 0–18 years, of all genders, who experienced any form of violence examined at the Paediatric Emergency Department of the ‘Maggiore della Carità’ Hospital in Novara (North-West Italy) between 1 January 2017 to 31 December 2021. Data were extrapolated by looking at the diagnosis at discharge. A comparison of the different variables collected was made between the pre-COVID-19 period and the COVID era. Results: 120 minors presented to the paediatric emergency room seeking help for violence. The average age was 10 years, 55% of the victims were male and 75% of them were Italian. In the pre-COVID period, the number of presentations for abuse was 62, while in the COVID period it was 58 with an increase of peer violence (from 38.71% to 62.07%) and with a statistically significant impact of the pandemic on the phenomenon (*p*-value < 0.00001). In general, peer violence accounts for 50% of the cases reviewed and resulted in fewer reports to the judicial authority and requests for forensic advice. Conclusion: The SARS-CoV-2-related pandemic has had an impact on total emergency room admissions and the types of abuse perpetrated.

## 1. Introduction

Child abuse is an endemic and underestimated problem that requires a multidisciplinary approach. Although it has been the topic of many studies, it is still difficult to describe and define due to limited availability of data internationally and different approaches to the problem.

Different ‘abuse’ concepts, used in 58 countries, were summarized in 1999 by the ‘World Health Organization’ (WHO) and subsequently investigated by various researchers and associations dealing with the phenomenon. Child maltreatment includes all forms of abuse and neglect affecting children under 18 years of age, such as physical (intentional use of force), emotional, psychological (i.e., isolation or exploitation), and sexual abuse (any sexual activity between an adult and a child who cannot legally give consent). Other forms of violence against minors are also neglect, exploitation for commercial or other purposes, resulting in real or potential damage to the health, survival, development or dignity of the child, and witnessing violence, a form of ‘indirect’ violence with the child forced to witness violent episodes against people to whom he or she is affectionately attached [1,2,3,4,5].

In addition to the forms of abuse mentioned above, bullying has recently been included among the forms of violence against minors. Bullying is characterised by violent and intimidating actions, and social isolation, against a child seen as a target by one or more peers. Alongside bullying, cyberbullying related to technologies in use by adolescents has developed. These two phenomena are the sign of poor integration toward those considered different by ethnicity, religion, or gender and can also have serious consequences (e.g., suicide attempt or suicide). Those involved in a relationship based on bullying have a three to five times greater risk of falling into suicidal thoughts: both victims and bullies in the six months following the violent incident have thought about self-harming. The relationship between bullying and suicide is not well defined, but a self-harm act carried out in the previous year can be considered a predictor of suicide. At the same time, being mentally ill and being a victim of other types of violence increases the risk of suicide in victims, while witnessing violence, physical abuse, and substance use increases the risk of suicide in bullies [6,7].

In 2006, WHO classified child maltreatment as a public health problem and, in 2020, reported that three out of four children between two and four years old regularly experienced physical punishment and psychological violence from their parents or caregivers; 120 million girls and young people under the age of 20 experienced some form of forced sexual violence, and there were an estimated 40,150 deaths per year from homicide in children under the age of 18 [8].

Internationally, Emergency Room (E.R.) admissions for abuse or maltreatment are between 2% and 10% of total admissions [9]. In Italy, in 2005, the rate of presentations for abuse in emergency departments was 2% [10].

The E.R. is often the entry point to healthcare for child victims of physical and sexual abuse, presenting with injuries such as bruises, abrasions, and fractures. So, the E.R. Department is an important checkpoint in the early recognition of suspected child abuse; unfortunately, clear management and diagnostic pathways have not been introduced on an international level yet [11].

In March 2020, the COVID-19 pandemic forced the entire population into social and physical distancing, with the closure of essential services. This led to forced prolonged cohabitation of adults and children, with an increased risk of violent episodes in dysfunctional families. According to the *Dossier Indifesa*, presented by Terres des Hommes in Italy during the lockdown period in 2020, there was a 13% increase in crimes of abuse against family members and cohabitants: 1260 girls and 1117 boys asked for police intervention [12].

Although an increase in abuse occurrence was expected, a decrease in Emergency Department presentations was observed due to the fear of contracting the virus. In this regard, the number of trauma presentations in Paediatric E.R. in the United States of America has undergone a change: analysing the period between 15 March and 15 May from the year 2015 to the year 2020, there was an expected general decrease in the number of presentations, and, during the pandemic period, there was a decrease in blunt trauma but an increase in burns and penetrating object-related injuries [13]. In Novara (North-West Italy), a 62% reduction in presentations to the Paediatric E.R. was observed between February and March 2020, compared to the same period in 2019: fewer children with minor severity codes were admitted [14].

Our study analyses the cases of child abuse examined at the Paediatric E.R. of ‘*Maggiore della Carità*’ Hospital in Novara, not only to take a picture of the violence against children and adolescents that occurred in that territory, but also to understand the gaps in the preparation of personnel who welcome and manage child victims of any form of violence and to introduce standardised procedures and create ad hoc training courses for health personnel. We considered the types of abuse and the characteristics of the victims, trying to identify possible hidden violence, discover the reason why it occurs and evaluate which characteristics are related to the presence of the injuries, the request for advice from specialists, and the reporting to the Judicial Authority. As secondary objectives, we are interested in assessing the change in the abuse prevalence during the study period, especially in the SARS-CoV2 era.

## 2. Materials and Methods

### 2.1. Ethical Approval

This study was submitted and approved by the Novara Intercompany Ethics Committee (CE162/2022).

Before submitting the project to the Ethics Committee, we considered the existing ethical structures for the use of data belonging to minors. We examined the question of consent to participate in the study and the legal framework for processing personal data reported in the medical records analysed. For further information on the discussion and the answers we have identified based on the legislation in force in our country, please refer to the Appendix A.

### 2.2. Setting

This was a cross-sectional study including minors, aged 0–18 years, of all genders, who had experienced some form of violence, asking for help for the presence of injuries or to report suspected abuse at the Paediatric Emergency Department of the ‘*Maggiore della Carità*’ Hospital in Novara (North-West Italy) between 1 January 2017 to 31 December 2021.

### 2.3. Data Collection

All data collected were evaluated based on the victim’s narrative or those accompanying the victim, such as family members or law enforcement.

Data were collected using the software “PS net” in place at the Paediatric Emergency Department. Data were extrapolated by two of the authors (MP and LR) independently by looking at the diagnosis at discharge most likely to identify suspected maltreatment such as “trauma,” concussion,” “other injuries,” “other causes of mortality and morbidity,” and in a few cases overt diagnoses of abuse were found. Then medical records were analysed to retrieve the information reported in the Appendix A (i.e., age of the victim, gender, nationality, type of maltreatment suffered, mode according to which the maltreatment occurs, relationship with the aggressor, injuries congruent with the story, use of objects at the time of the abuse, legal procedures undertaken by health professionals, days of prognosis, need for hospital admission, other advice required for the valuation of the maltreatment—Appendix A.

### 2.4. Statistical Analysis

Descriptive statistics were used to summarise the characteristics of the included subjects: categorical variables were reported as absolute frequencies and percentages while numerical variables were reported as mean and standard deviation (SD) or median and first (Q1) and third quartile (Q3) if not normally distributed according to the Shapiro—Wilks test. The association between the patient’s characteristics and reporting to the judicial authority, the presence of injuries, and seeking forensic advice was tested using the chi-square or Fisher exact test for categorical variables and the t-test or Mann—Whitney test for numerical variables. Moreover, Poisson regression models with robust variance were fitted to estimate the prevalence relative risk (pRR) and the corresponding 95% confidence intervals (95% CI) for the association between the patient’s characteristics and reporting to the judicial authority, the presence of injuries and seeking forensic advice. The tests were two tailed and the type I error was set to 0.05. Statistical analyses were performed using SAS version 9.4 (SAS Institute, Cary, NC, USA).

## 3. Results

Child abuse admissions to the Paediatric Emergency Department were studied from 1 January 2017 to 31 December 2021. Figure 1 reports the total presentations to the E.R. by year and the number of minors examined for any form of abuse.

Overall, 120 cases of abuse out of 60,622 presentations to the Paediatric E.R. were identified by emergency room health personnel between 2017 and 2021, corresponding to a prevalence 0.20%. A steady increase in the prevalence of abuse was observed, specifically, it varies from 0.13% in 2017 to 0.34% in 2021. Even in the two years of the pandemic, despite a reduction in hospital admissions, there was an increase in the percentage of cases of violence against minors identified by health personnel and/or reported by the minors themselves or by those who assisted them.

### 3.1. General Characteristics of Minors

The general characteristics of the minors examined at the E.R. in Novara are summarised in the Table 1.

Minors who were examined for suspected abuse were males in 55% of cases and had an average age of 10 years. Two of the children involved in suspected violence cases had mental deficits: one of the minors had a behaviour disorder with mild cognitive retardation, while the second had mild mental retardation.

A possible relationship between the types of maltreatment and age can be assessed. The cases were grouped into age groups (Figure 2): 1–5 years, 6–10 years, 11–18 years. It was found that the most affected group for both physical abuse and peer violence is 11–18 years old.

Regarding sexual abuse, both the youngest (1–5 years) and adolescents (11–18 years) appear to be the victims with the highest incidence. Neglect and witnessing violence occur only in the lower age groups. However, it is worth mentioning that, given the paucity of data, it is not possible to draw statistically reliable conclusions.

Most of the patients involved were Italian (75%). Figure 3 shows the distribution of the 30 foreign children according to their area of origin. Most of the victims were of African origin (56.67%), while the least frequent origin is Western Europe.

### 3.2. Types of Abuse

50.00% of all cases analysed are attributable to peer abuse, 37.50% to physical abuse. Then, in 10.00% of the cases, minors were victims of sexual violence; finally, witnessing violence and negligence appear respectively in 1.67% and 0.83% of the events (Figure 4).

### 3.3. Maltreatment and Sexual Abuse in the Pre-COVID and COVID Period

To understand the impact of the COVID pandemic on child abuse, a comparison was made on the type of violence in the pre-COVID and COVID-19 period. In the pre-COVID era, 45.16% of E.R. visits related to violence were for mistreatment and 11.29% for sexual abuse, while in the pandemic they were respectively the 29.31% and 8.62% of the recorded violence. For these two types of abuse, the SARS-CoV-2 pandemic did not have a statistically significant impact (*p*-value = 0.398 for sexual abuse, *p*-value = 0.2846 for physical abuse), but considering the E.R. admissions for peer violence, they increased from 38.71% in the pre-COVID period to 62.07% in the pandemic-related era, with a statistically significant impact of the pandemic on the phenomenon (*p*-value < 0.00001).

### 3.4. Relationship between Victim and Aggressor

In the 120 cases observed, 30.83% of the violence was caused by a person of the same age, 23.33% by a family member, 18.33% involved more people (becoming group violence), 15.83% involved an unfamiliar adult, and for 11.67% of the events no data on the aggressor was found in the medical record.

### 3.5. Injuries

Following the medical procedure, anamnesis and a medical examination were carried out in cases of suspected violence to determine the presence of injuries. In 77.50% of cases, we have evidence of lesions that were suggestive of some form of violence.

In 12.50% of cases, the injuries were inflicted using different types of objects, such as cell phones, remote controls, electric cables, and rolling pins. This information was gathered through the history and narrative of the child victim of violence.

### 3.6. Specialist Advice

Most of the specialist advice was requested for the management of sexual violence episodes. Approximately 14% of these forms of abuse required the intervention of a gynaecologist or forensic doctor.

### 3.7. Prognosis

After evaluating patients with a suspected diagnosis of violence, based on the child’s medical history and health, the correct hospital course of action was decided: 94.07% of children were discharged, and in a few cases of these, an outpatient course was implemented. Only 5.93% of survivors were hospitalized.

Following the discharge diagnoses, prognosis days were determined: the median number of days was two.

### 3.8. Child Abuse and Legal Procedures Undertaken by Health Professionals

#### 3.8.1. Art. 403 Italian Civil Code

This article provides for the removal and protection of the minor morally or materially abandoned or who is in physical and psychological danger. This measure allows the child to be safeguarded from potentially harmful situations in the family environment that can cause an emergency in the child’s life. It can be issued during the medical examination in the emergency room or during hospitalization when health workers have a strong suspicion of family mistreatment or abuse. In our sample, in only one case of those identified (0.83%), it was activated because the history of this minor was already known by social services.

#### 3.8.2. Reporting to the Judicial Authorities

In this study, 61.67% of suspected cases of child abuse were reported to the judicial authorities by the health professional.

#### 3.8.3. Distribution of Subject’s Characteristics According to the Event

Table 2 shows the distribution of the subject’s characteristics according to whether the event was reported to the judicial authority, the presence of injuries and if forensic advice was sought, the *p*-value of the test used to evaluate the association between the patient’s characteristics and these events, and the pRR and corresponding 95% CI.

The type of abuse (*p*-values: 0.0286 for reporting to the judicial authority, 0.0028 for presence of injuries and <0.0001 for seeking forensic advice) and the relationship between victim and aggressor (*p*-values: 0.0002, 0.0114, 0.0005) were associated with all events considered. When the abuse reported was peer violence, it was less likely to be reported to the judicial authority (pRR 0.68, 95% CI 0.51–0.92) or that forensic advice was sought (pRR 0.09, 95% CI 0.01–0.72) compared to physical abuse, while sexual abuse was less likely to cause injuries compared to physical abuse (pRR 0.49, 95% CI 0.25–0.98). Regarding the relationship between victim and aggressor, compared to abuse perpetrated by a peer, violence committed by a relative or other adults leads to a probability of reporting to the judicial authority 2.31 and 2.43 times higher, respectively, as well as an increased probability of requiring forensic advice (pRR 5.59 and 5.84 respectively). Abuse committed by adults was less likely to cause injuries compared to those committed by peers (pRR 0.63, 95% CI 0.42–0.94). Group violence was associated with an increased probability of reporting to judicial authority compared to peer violence. Moreover, sex was associated to both injuries and a need for forensic advice, specifically, males were less likely than females to be subjected to injuries (pRR 0.81, 95% CI 0.67–0.99) while females were more likely to need forensic advice (pRR 2.93, 95% CI 1.10–7.81). Finally, the number of prognosis days was associated with reporting to the judicial authority, increasing the prognosis by one day led to an increase in a report’s probability of 1.06.

## 4. Discussion

As already discussed in the “Introduction” paragraph, the Emergency Department is often the entry point to health care for survivors of any form of violence, regardless of age or gender. In our study, we recorded a total of 120 presentations for violence against minors at the Paediatric Emergency Department of *Maggiore della Carità Hospital* in Novara out of a total of 60,622 presentations in the period between 2017 and 2021. Approximately 0.19% of minors who were visited at the hospital were maltreated. Our data differ from the international, national, and regional reality described in the report of the REVAMP project (*REpellere Vulnera Ad Mulierem et Puerum* project) [9], promoted in Italy by the *Istituto Superiore di Sanità* between 2014 and 2017.

Furthermore, the average number of presentations to the Novara Paediatric E.R. is 24 cases/year, less than the 82 suspected cases/year observed in the E.R. of the ‘Bambino Gesù’ Hospital in Rome during the period from January 2008 to October 2020 [15]. This decrease could be due to the different population density in Novara compared to Rome, but also to the introduction of an early form of management applied to all minors examined at the Roman hospital. This form evaluates three variables that can be linked to violence (incongruent anamnestic statements, neglect, and injuries found at physical examinations) and an increase was observed in the number of cases recognised as violence (82 cases/year) compared to the number recorded, when this approach was not in force.

Other realities have also highlighted the importance of introducing screening tools: in Amsterdam between 2011 and 2013, it was noted how combining screening tests and more thorough physical objective examinations led to an increase in child abuse diagnoses at the Emergency Department [16]. To date, as highlighted in the REVAMP project report, a single screening tool shared by the international scientific community has not yet been created. However, in various studies, researchers have codified indicators, measuring their sensitivity and specificity. Among all currently validated screening systems, the one that seems to have the simplest application, good sensitivity and high specificity resulted to be the list of questions—indicators of the ESCAPE project proposed by Louwers in 2014 [17].

One limitation of our study is that we only considered episodes of violence that required medical intervention. This does not allow us to fully understand the spread of the phenomenon. There may be episodes of child abuse that remain hidden because, even if they happen in public places (e.g., parks, schools...), the victims are subjugated by their attackers, or because they do not result in serious damage to the quality of life of children or occur in family contexts.

Regarding the evaluation of the data obtained in relation to the period in which the episodes of violence occurred, our study shows how the COVID-19 pandemic had a statistically significant impact on child abuse in the Province of Novara. In fact, 58 cases of abuse were recorded on 18,295 admissions to the emergency room compared to 62 cases on 42,327 admissions in the pre-COVID era.

The types of maltreatment against minors who went to the Emergency Room were studied: assessing the distribution of the types of maltreatment concerning the cases considered, the evolution of peer violence is evident. In 2020, ISTAT showed that more than 50% of the surveyed minors, aged 11 to 17, reported having been bullied in the previous 12 months [18]: technological changes, social transformations, and economic growth have encouraged the growth of this phenomenon among adolescents, in which superiority of a peer group or a single perpetrator against a victim is practiced. In our research, the COVID-19 pandemic had a significant impact on peer violence. This figure might seem to contrast with the isolation imposed by the pandemic, but in Italy periods of total closure have alternated with periods in which social life has resumed for both adults and children. In our study, the cases of peer violence mainly occurred with the return to school: it was precisely the school that was the place where it happened most frequently. In our sample, no cases of cyberbullying were recorded, even if the literature highlighted an increase during the pandemic period due to the misuse of electronic devices and social networks [19], often associated with poor supervision by parents and the loss of support programs from associations and social services.

In general, in our territory, as well as globally, during the pandemic period, due to social distancing and the interruption of services, far fewer children went to the hospital for help and some incidents remained unacknowledged. However, it emerges from the literature that in this historical moment some types of abuse have instead increased due to the stress of parents for the daily management of their children, the loss of their job, the closure of commercial activities and essential services (e.g., schools), the decrease of preventive actions (e.g., of prevention such as home visits of health personnel to support the growth of children, programs conducted in schools aimed at children) [2,13,20,21,22,23,24,25].

In 2020, Rapoport conducted a study estimating the number of abuses in New York City, comparing the pre-COVID period to the COVID period reported in March, April, and May 2020: a 29% decrease was observed during the pandemic [26]. This is in line, not only with what has been described by other authors regarding the situation in different US states [27] and in France [20], but also with the results of our study. There was a 6% reduction in reports at E.R. in Novara during the COVID period.

Considering the results obtained in relation to the type of aggressor and regardless of the period, our study showed that the abuser was in most cases a peer. In Thailand between the years 2002 and 2017, 133 cases of child maltreatment were identified in a children’s hospital and 45.10% of these cases were supported by parents. It was also shown that the presence of abusive relatives and age between one- and 10-years old lead to a higher risk of recidivism [28]. No abuse with recidivism was identified in our study, except for one case of family maltreatment already known to the competent authorities.

In our sample, the average age of the juvenile population was 9.92 years, while in Spain between 2007 and 2018 children involved in maltreatment have an average age of 4.3 years [29]. This increase in the average age could be related to the fact that in Novara there were more episodes of peer violence than physical, sexual abuse, neglect, or carelessness, which are typical of early and later childhood.

In our study, the Italian victims are 90 children out of 12,890 Italian minors residing in Novara; as far as foreign nationalities are concerned, we highlight a low incidence in our area (30 out of 3286 foreign children): in these cases, a lot of mistreatment goes unreported and remains hidden, and this is the reason why a low presence of foreign victims is reported.

Most of the maltreatment cases studied do not have days of prognosis indicated where the average is two days, and prognoses with more days are associated with diagnoses of physical abuse by peers or family members. Almost all minors were discharged home after completion of the E.R. clinical course: only 10 out of 120 minors with suspected maltreatment diagnoses were hospitalized. The hospitalizations are linked to diagnoses of physical abuse within the family or sexual abuse. This can be considered “favourable”, in that the physical injuries inflicted were not considered disabling in the child’s life, but it makes us pose a consideration of how much, once the medical process is over, the child is not followed in daily life and may favour repeated episodes of violence. In 99.17% of cases, child protection services (article 403 C.C.) were not activated even though some situations would have deserved more attention to protect victims. Moreover, it was shown that a forensic or gynaecological consultation was required to ascertain whether there were physical signs that could suggest a violence in only 14.17% of the cases. This advice proves to be very useful when the story told by the minor or the parents is not clear or is not compatible with the injuries sustained.

The infrequent activation of child protection services and the low number of specialist consultations required in our reality underline the need to train healthcare staff more in a multidisciplinary approach in the management of victims of violence, whatever the gender and age of the person involved. Furthermore, a multidisciplinary approach is recommended to propose a personalized pathway appropriate to the event of which the child was a victim.

When the child goes to the Emergency Room seeking help, the presence of signs is assessed: it correlates with physical maltreatment and peer violence, showing a marked difference from the evidence of physical consequences of sexual abuse. This finding is almost in line with a study published in 2010 by Ermertcan, in which hematomas, lacerations, and abrasions are observed in 90 percent of physical abuse cases [30]. The presence or absence of injuries cannot be considered a predominant factor in making a diagnosis of maltreatment: the necessary pillar for a valid pathway and help for the child is the medical history. The child needs to feel comfortable, not judged but understood. Creating a safe, comfortable, and trusting environment allows the spontaneous storytelling of the victim, who can be supported in the best way possible. For this reason, the involvement of multiple professionals to coordinate and safeguard the protection and growth of the child is essential.

Better management of abuse survivors in the Emergency Department, whatever their age and gender, would be possible with the introduction of specific protocols and the training of health personnel as suggested by different authors in the literature [31,32,33,34,35,36]. In Italy, there are few health centres dedicated to the management of minors who suffer any form of abuse (e.g., Milan, Turin). For these reasons, Italian health professionals increasingly feel the need to have shared protocols and specific training available. This is necessary to be able to receive and treat young victims in the most appropriate way, to send only the most complex patients or those in whom the suspicion of abuse is very nuanced to the dedicated centre, which is often located hundreds of kilometres away. In 2010, the Ministry of Health published the volume “*L’abuso sessuale nei bambini prepuberi–Requisiti e raccomandazioni per una valutazione appropriata*” (“*Sexual abuse in prepubescent children. Requirements and recommendations for an appropriate evaluation*”) [37]. The document, aimed at all professionals who work with minors, offers various tools for a correct approach to minors with suspicion of sexual abuse: a diagnostic path, a model of a medical record and a complaint, and a complaint treatment to the judicial authority. Over the years, the Italian regions have acted individually with regional guidelines for the care of abused children and adolescents. However, there is still a lack of national guidelines for healthcare and hospital companies about the rescue and social—health assistance for child victims of abuse, along the lines of what has been issued for women victims of violence in 2017 [38].

Approved guidelines are essential to improve medical practice, especially in the management of specific situations, even if the standardization of patient management is not always possible, since as stated by Donati et al. in 2020, specific conditions often require adjustments to the usual standard of care. The same authors underline how the guidelines aim to provide a brief instruction on how to provide up-to-date health services compliant with the latest advances, assisting health personnel in the decision-making process, in order to give the patient better clinical care, and in case specific also medical examiner [39].

At the same time, the production of guidelines should be associated with specific training for health professionals, not only emergency workers, but also, for example, family paediatricians and nurses. Training aims at recognizing the signs and symptoms that may indicate possible child abuse. It is necessary to create in the healthcare staff the awareness that abuse should be considered as one of the possible differential diagnoses. In clinical practice, for example, the differential diagnosis between Abusive Head Trauma (AHT) and other similar conditions (e.g., accidental head trauma, hypoxic-ischemic injury) is a common dilemma not only from a clinical point of view, but also from a forensic aspect because the seriousness of the consequences of an erroneous diagnosis (false positive or false negative) requires the definition of precise diagnostic criteria. It is important to know that some signs and symptoms, while non-specific, have turned out to be “red flags” for abuse. Healthcare personnel should know that after examining the child, laboratory tests and imaging studies are needed. The results of each step performed for the diagnosis must be evaluated as a whole to better identify the misunderstood cases of maltreatment [40].

In addition, healthcare personnel must be trained on how to complete the medical record. It is necessary to accurately report the physical and psychological state of survivors. In fact, according to the literature, even if the mere presence of medical documentation, without the support of other sources of evidence, is often not definitive for the verdict, in cases where there are multiple sources of evidence, the clinical documentation can provide useful elements. It may in fact indicate the consistency between the reported history and the observed lesions. A greater number of convictions are recorded in cases in which the victim has produced clinical documentation regardless of whether the victim has presented to the General Emergency Department or to a specific anti-violence service, suggesting the importance that the medical documentation has. [41]

## 5. Conclusions

The research carried out made it possible to evaluate the phenomenon of child abuse in the province of Novara, using presentations to the Paediatric Emergency Room as a tool. Despite the lack of some information, we can underline that the prevalence of violence in Novara is lower than in the Italian territory. The COVID-19 pandemic has had a significant impact on the diagnosis of abuse and the type of violence, recording an increase in peer violence in our area. Some deficiencies in the management of survivors have emerged: the infrequent activation of child protection services and the low number of specialist consultations required. This underlines the need for greater training of healthcare personnel in a multidisciplinary approach.

The continuous training of health personnel in the management of this type of emergency becomes essential. It is also essential to follow specific and targeted paths to identify the phenomenon, to involve social operators in accompanying the families of the victims, and to introduce a broader coordination between the network, which not only includes the health personnel, the social assistants, but also the judicial authority and the school.

All these coordinated actions can contribute to the creation of a support and safety network for minors. Lifelong assistance to the child should be ensured in the form of health, employment, and social assistance so that the child can integrate into the community and continue to grow.

## Figures and Tables

**Figure 1 ijerph-20-02028-f001:**
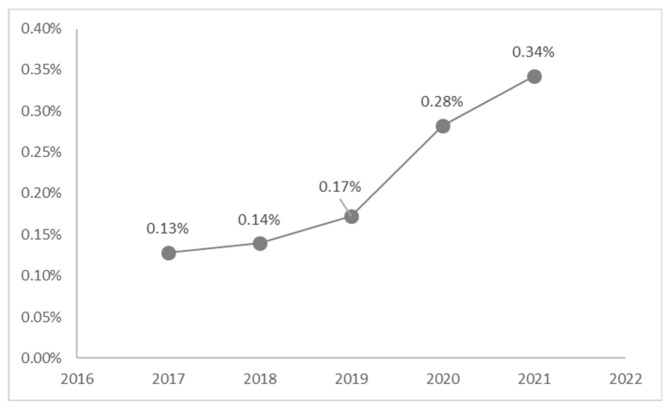
Represents the cases of abuse related to the total number of Paediatrics E.R. admission by year.

**Figure 2 ijerph-20-02028-f002:**
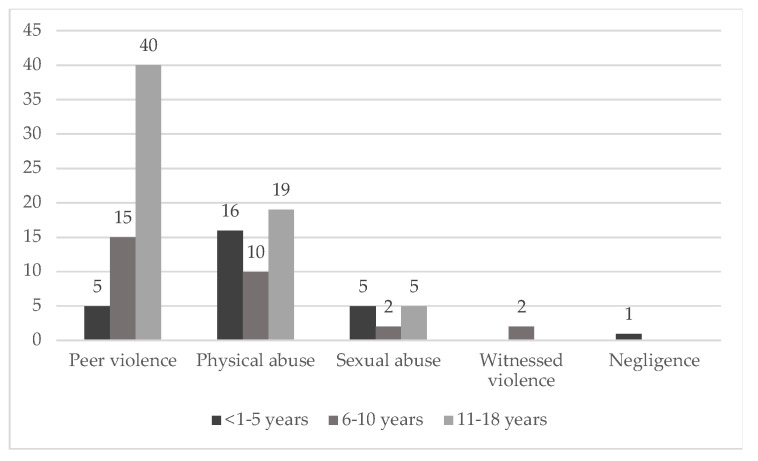
Distribution of abuse by age group.

**Figure 3 ijerph-20-02028-f003:**
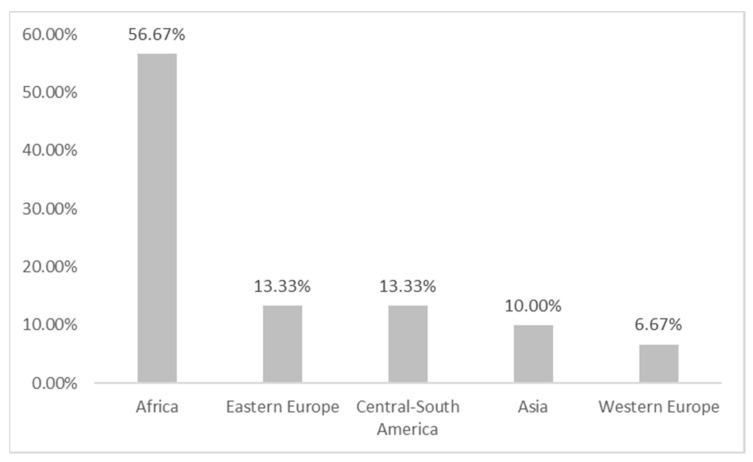
Distribution of foreign minors according to the country of origin.

**Figure 4 ijerph-20-02028-f004:**
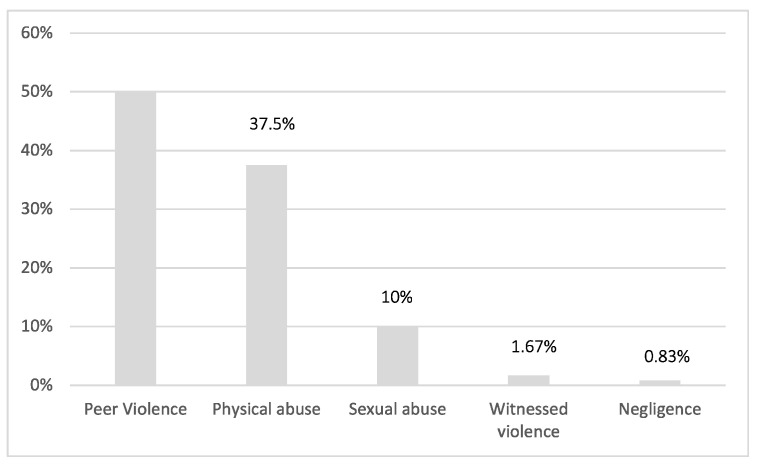
Distribution of cases of mistreatment.

**Table 1 ijerph-20-02028-t001:** Distribution of the minor’s characteristics.

	N = 120
**Variable**	N (%)
**Sex**	
F	54 (45.00)
M	66 (55.00)
**Age, mean (SD)**	9.92 (4.54)
**Time period**	
COVID	58 (48.33)
Pre-COVID	62 (51.67)
**Year**	
2017	18 (15.00)
2018	20 (16.67)
2019	24 (20.00)
2020	22 (18.33)
2021	36 (30.00)
**Origin**	
Italian	90 (75.00)
Foreign	30 (25.00)
**Type of abuse**	
Physical abuse	45 (37.50)
Sexual abuse	12 (10.00)
Negligence	1 (0.83)
Witnessed violence	2 (1.67)
Peer violence	60 (50.00)
**Abuser**	
Peer	37 (30.84)
Relative	28 (23.33)
Adult (not relative)	19 (15.83)
Not reported	14 (11.67)
Group violence	22 (18.33)
**Reported use of objects in abuse**	
No	105 (87.50)
Yes	15 (12.50)
**Physical or mental deficit**	
No	118 (98.33)
Yes	2 (1.67)
**Child protection services (403 Activation)**	
No	119 (99.17)
Yes	1 (0.83)
**Report to the judicial authority**	
No	46 (38.33)
Yes	74 (61.67)
**Forensic advice sought**	
No	103 (85.83)
Yes	17 (14.17)
**Presence of injuries**	
No	27 (22.5)
Yes	93 (77.5)
**Need for hospital admission**	
No	111 (92.50)
Yes	7 (5.83)
Missing	2 (1.67)
**Prognosis (days) median, Q1–Q3**	2 (0–3)

**Table 2 ijerph-20-02028-t002:** Distribution of subject’s characteristics according to the event, *p*-value of the association test, pRR and corresponding 95% CI.

	Report to the Judicial Authority				Presence of Injuries				Forensic Advice			
	No N = 46	Yes N = 74	*p*-Value	pRR (95% CI)	No N = 27	Yes N = 93	*p*-Value	pRR (95% CI)	No N = 103	Yes N = 17	*p*-Value	pRR (95% CI)
Variable	N (%)	N (%)			N (%)	N (%)			N (%)	N (%)		
**Sex**												
F	20 (43.48)	34 (45.95)	0.7916	1	17 (62.96)	37 (39.78)	0.0331	1	42 (40.78)	12 (70.59)	0.0221	1
M	26 (56.52)	40 (54.05)		1.04 (0.78–1.38)	10 (37.04)	56 (60.22)		0.81 (0.67–0.99)	61 (59.22)	5 (29.41)		2.93 (1.10–7.81)
**Age, mean (SD)**	9.24 (4.64)	10.4 (4.45)	0.1984	1.02 (0.99–1.06)	8.81 (4.95)	10.24 (4.39)	0.1525	1.02 (0.99–1.04)	10.11 (4.89)	8.76 (4.79)	0.2602	0.95 (0.87–1.04)
**Time period**												
COVID	22 (47.83)	36 (48.65)	0.9301	1	14 (51.85)	44 (47.31)	0.6777	1	47 (45.63)	11 (64.71)	0.1448	1
Pre-COVID	24 (52.17)	38 (51.35)		1.01 (0.76–1.34)	13 (48.15)	49 (52.69)		0.96 (0.79–1.17)	56 (54.37)	6 (35.29)		1.96 (0.78–4.96)
**Year**												
2017	10 (21.74)	8 (10.81)	0.4534	1	6 (22.22)	12 (12.9)	0.4799 *	1	14 (13.59)	4 (23.53)	0.3524 ^§^	1^§^
2018	6 (13.04)	14 (18.92)		1.58 (0.87–2.84)	3 (11.11)	17 (18.28)		1.28 (0.88–1.86)	18 (17.48)	2 (11.76)		0.45 (0.09–2.17)
2019	8 (17.39)	16 (21.62)		1.50 (0.83–2.70)	4 (14.81)	20 (21.51)		1.25 (0.86–1.81)	24 (23.3)	0 (0.00)		Not estimable
2020	7 (15.22)	15 (20.27)		1.53 (0.85–2.77)	7 (25.93)	15 (16.13)		1.02 (0.66–1.58)	20 (19.42)	2 (11.76)		0.41 (0.08–1.98)
2021	15 (32.61)	21 (28.38)		1.31 (0.73–2.36)	7 (25.93)	29 (31.18)		1.21 (0.84–1.74)	27 (26.21)	9 (52.94)		1.13 (0.40–3.16)
**Origin**												
Italian	35 (76.09)	55 (74.32)	0.8284	1	19 (70.37)	71 (76.34)	0.528	1	80 (77.67)	10 (58.82)	0.1291*	1
Foreign	11 (23.91)	19 (25.68)		1.04 (0.75–1.43)	8 (29.63)	22 (23.66)		0.93(0.73–1.18)	23 (22.33)	7 (41.18)		2.10 (0.88–5.03)
**Type of abuse**												
Physical abuse	11 (25.58)	34 (45.95)	0.0286	1	7 (29.17)	38 (40.86)	0.0028	1	37 (37)	8 (47.06)	<0.0001	1
Sexual abuse	3 (6.98)	9 (12.16)		0.99 (0.69–1.43)	7 (29.17)	5 (5.38)		0.49(0.25–0.98)	4 (4)	8 (47.06)		3.75 (1.78–7.90)
Peer violence	29 (67.44)	31 (41.89)		0.68 (0.51–0.92)	10 (41.67)	50 (53.76)		0.99 (0.83–1.17)	59 (59)	1 (5.88)		0.09 (0.01–0.72)
**Relationship between victim and aggressor**												
Peer	25 (54.35)	12 (16.22)	0.0002	1	3 (11.11)	34 (36.56)	0.0114 *	1	35 (33.98)	2 (11.76)	0.0005 ^§^	1 ^§^
Relative	7 (15.22)	21 (28.38)		2.31 (1.39–3.86)	4 (14.81)	24 (25.81)		0.93 (0.78–1.12)	20 (19.42)	8 (47.06)		5.29 (1.22–22.98)
Adult (not relative)	4 (8.7)	15 (20.27)		2.43 (1.45–4.09)	8 (29.63)	11 (11.83)		0.63 (0.42–0.94)	13 (12.62)	6 (35.29)		5.84 (1.30–26.23)
Not reported	6 (13.04)	8 (10.81)		1.76 (0.92–3.37)	5 (18.52)	9 (9.68)		0.70 (0.47–1.05)	14 (13.59)	0 (0.00)		Not estimable
Group violence	4 (8.7)	18 (24.32)		2.52 (1.52–4.18)	7 (25.93)	15 (16.13)		0.74 (0.55–1.00)	21 (20.39)	1 (5.88)		0.84 (0.08–8.75)
**Use of objects**												
No	39 (84.78)	66 (89.19)	0.4779	1					90 (87.38)	15 (88.24)	1.0000 *	1
Yes	7 (15.22)	8 (10.81)		0.85 (0.52–1.39)					13 (12.62)	2 (11.76)		0.93 (0.24–3.68)
**Prognosis (days) median, Q1–Q3**	0.5 (0–2)	3 (0–4)	0.0013^	1.06 (1.01–1.11)	0 (0–3)	2 (0–3)	0.0238^	1.02 (0.98–1.07)	2 (0–3)	2 (0–3)	0.9168 ^	0.98 (0.84–1.15)

^ Mann–Whitney test, * Fisher exact test, ^§^ category with event frequency 0 was excluded from the analysis.

## Data Availability

Not applicable.

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
