# Peer review of "Clinical and Forensic Aspects of the Management of Child Abuse: The Experience of the Paediatric Emergency Department in Novara, North-West Italy"

_ijerph, 2023, doi:10.3390/ijerph20032028_

Round 1

Reviewer 1 Report

In analysing your paper, I came across some interesting points that would also be worth examining in more detail

1) I believe that a more in-depth investigation is needed by comparing the current literature to present the current state of scientific knowledge; I think the following articles in particular could be useful:
Maiese et al. Pediatric Abusive Head Trauma: A Systematic Review
Donati et al. A Perspective on Management of Limb Fractures in Obese Children: Is It Time for Dedicated Guidelines?
Blandino et al. Sexual assault and abuse committed against family members: An analysis of 1342 legal outcomes and their motivations

2) How do you explain the increase in peer violence despite the closure of schools and the reduction in contact between children?

3) I think it would be helpful to better specify the reasons for visiting ER?

4) I think it would be helpful to better specify the selection criteria used to select suspected cases of abuse or violence from the database (was a questionnaire used?) 

Reviewer 2 Report

Important topic, excellent consideration of key factors about abuse cases coming to the ER. Implications of the findings are impactful to improving care for children and reducing repeated and life-threatening harms, which ultimately improves children’s outcomes. This matters to long-term outcomes for individuals, families, and the broader public health for a community. Recommendations below are intended to clarify important points in the paper and strengthen the contribution to the literature.

Abstract:

For Methods summary, I would remove first portion “several parameters….” And start with “Comparisons were made between X, X, and X (Xs to be replaced with the key parameters) in pre-COVID and pandemic time periods.” State types of statistical tests used.

For Results summary, recommend providing the pre/post on each demographic and occurrence variable of importance and the statistical difference/not difference and stat used to compare for differences.

For Conclusion summary, would highlight more the distinct pattern difference and potential impact on care (more specifics).

Overall, recommend the abstract be a stronger representation of the specific work that has been done with key areas of results and conclusion explicated so readers can appreciate the importance of the analysis and impact on care.

Intro:

Solid overview and definition of types and prevalence of child abuse. Would restate start of 4th paragraph, page 2 to be more clear (currently sounds like ER is only option instead of a frequent access point): The E.R. is often the access point to healthcare for child victims of physical abuse, presenting with injuries such as bruises, abrasions, and fractures. Page 2, paragraph 6: restate to “forced prolonged cohabitation” to acknowledge the time element vs. the usual living circumstance. Page 2, paragraph 7: recommend “an increase in abuse occurrence was expected”

Methods:

Well described and appear appropriate to plan.

Results:

On Table 1, would describe a couple of the variables more so table is able to be understood on its own:

Ie Use of Objects – Reported use of objects in abuse, Activation 403 -Child protection services (403 Activation), Forensic advice sought, Physical or mental deficit

Discussion:

Nicely laid out summary of the different areas and bringing together to the greater importance in the findings.

In paragraphs 4, 5, and 6, would focus more on what the results indicate and make any additional conjectures (effects of social media, etc.) with additional literature back up.

In paragraph “A better management of the minors….”, Increase discussion/add literature back up for some current protocols in place elsewhere or what evidence offers. This is such a critical point in your overall results and their impact on care and outcomes-develop it further.

Conclusions:

Ensure the bullet areas here have been well explored and specific strategies proposed in the Discussion area for each. Very relevant points on what is needed are provided-be sure the article is bringing together existing knowledge/strategies that have been tested to build the evidence base towards specific actions being taken.

General/Language/Style:

Some points are made with 1 or 2 sentence segments; evaluate how to reduce length of longer sentences to have more defined thoughts without long sentences that can be hard to manage for the reader. Collect points into 3 sentence minimum paragraphs to allow better flow for the reader and development/linking of points.

An additional review for English language use/translation recommended across whole manuscript for minor rewording and correction of usual English order of statements. Examples: page 4 under 3.1: Most of the children involved in suspected violence cases did not have specific health needs, with only 1.67% having an identified physical or mental difference. Example-Results 3.6: Additional counseling was needed for all sexual assaults. (Also, clarify this statement with the next (explain further difference in counseling vs. forensic or gynecological medical advice). Reword recommended for last paragraph, page 11 (“and get rid of it” too casual).

Reviewer 3 Report

The article is interesting and addresses a phenomenon not much dealt with in the field of clinical forensic medicine.

However, the text is quite generic and needs more focusing on the aim of the research. Some inconsistencies need to be reviewed.

The title is misleading: why do the authors discuss the north-west of Italy taking a single hospital into account? Please, change the title also adding the name of the city.

Introduction: this part is excessively long and verbose. Some parts could be added directly into the discussion, others removed as unnecessary.

A more in-depth investigation of the current literature, especially the national and European one, is strongly needed. The current state of scientific knowledge is fundamental and such a small reference weakens the whole manuscript.

Here some suggestions, some of them also dealing with pandemic issues in France:

-Prigent A, Vinet MA, Michel M, Rozé M, Riquin E, Duverger P, Rousseau D, Chevreul K. The cost of child abuse and neglect in France: The case of children in placement before their fourth birthday. Child Abuse Negl. 2021 Aug;118:105129. doi: 10.1016/j.chiabu.2021.105129.

- Hébert M, Smith K, Caouette J, Cénat JM, Karray A, Cartierre N, Veuillet-Combier C, Mazoyer AV, Derivois D. Prevalence and associated mental health outcomes of child sexual abuse in youth in France: Observations from a convenience sample. J Affect Disord. 2021 Mar 1;282:820-828. doi: 10.1016/j.jad.2020.12.100.

- Caron F, Plancq MC, Tourneux P, Gouron R, Klein C. Was child abuse underdetected during the COVID-19 lockdown? Arch Pediatr. 2020 Oct;27(7):399-400. doi: 10.1016/j.arcped.2020.07.010.

- Caron F, Tourneux P, Tchidjou HK, Taleb A, Gouron R, Panuel M, Klein C. Incidence of child abuse with subdural hemorrhage during the first year of the COVID-19 pandemic: a nationwide study in France. Eur J Pediatr. 2022 Jun;181(6):2433-2438. doi: 10.1007/s00431-022-04387-x.

It would also be interesting to have a better understanding of how the management and care of children in the health care system takes place on a procedural level in Italy and if there are some national centres, with different protocols, perhaps as in Turin, Milan or Florence.

Round 2

Reviewer 1 Report

I find that the changes made to the article have improved it. Therefore, having developed the potential of the study and deepened its shortcomings, I believe that it makes an important contribution to the relevant literature at this stage.

Reviewer 3 Report

In my opinion, the manuscript now is worth of publication. Congratulations.